

# Travel Sensei
## Turning Travel into Adventure

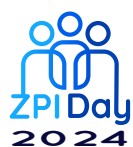

**Authors**: Valentina Bolbas ⓘ · Maurycy Jakiel ⓘ · Daniil Kuznetsov ⓘ · Abrorjon Ruziboev ⓘ · Volodymyr Shepel ⓘ

**Supervisor:** Wojciech Thomas

### Abstract

The Travel Sensei project is designed to make travel more accessible, enjoyable, and engaging, using gamification, social interaction, and personalisation. This mobile application encourages active travel exploration, ignites a sense of adventure, and at the same time promotes mental health and strengthens community ties. The application reinvents travel as a socially rewarding and competitive activity by integrating features such as interactive maps, achievement-based milestones, and a robust friendship system. Aesthetically pleasing design and personalised user profiles ensure a seamless and enjoyable experience, enabling individuals to express their unique travel identities.

Focused on community-building and enhancing the travel experience, Travel Sensei combines fun, accessibility, and connection to create a platform that turns travel into an adventure worth sharing.

## 1 DEVELOPMENT

### 1.1 Introduction

With the Travel Sensei project, we want to solve the problem of travel lacking engagement and social connection, making it hard for people to stay motivated and share meaningful experiences while exploring new places. Many people spend most of their time indoors and rarely go outside without a specific and clearly defined goal. Our primary objective with the Travel Sensei project is to motivate people to go and explore their surroundings with true enthusiasm and involvement. The crucial part for us is to promote mental and physical well-being, as well as to support tourism by encouraging exploration of various locations.

To solve the problem defined above, we set out to develop a mobile application. The main goals of this project are:

- develop gamification elements to reward exploration and engagement;

- create a consistent and aestetically pleasing user interface;

- introduce a social platform for travellers to share and connect;

- foster a community around shared travel experiences;

- provide a smooth user experience while using our application.

We are confident in our approach and the potential to combine aspects of game design with travel and tourism. The business benefits that we want to highlight with the Travel Sensei are the following:

- increase in user engagement, achieved via interactive features like an achievement-based map, badges, and leader boards;

- high level of personalisation, included through unified themes and customizable user profiles, which help users to express individuality to others and stand out from the crowd;

- creation of a community built on a friendship system, which serves to build strong connections while stirring up the competition in an affable manner.

## 1.2   Related Work

Travel Sensei introduces a novel approach to travel-focused applications by integrating engagement-driven features that transform travelling into a social, competitive, and entertaining experience. With the application, we place a great deal of emphasis on "gamification" as a way to boost user engagement, enjoyment, and usage. As was shown in the example of *Pokemon GO*, which "gamified" tourist exploration, this approach shows a measurable increase in exploration carried out by travellers in new locations, as well as increased happiness and well-being [11]. Our project tries to do the same, by introducing game elements, such as collecting points, getting milestones, climbing leader-boards and more. To contextualize its innovation, this section reviews existing solutions and technologies in the domain and highlights how Travel Sensei stands out.

### 1.2.1   Competitor Analysis

- *Google Maps* provides a robust travel utility with its navigation, discovery, and location-marking features. However, it lacks gamification elements that engage users beyond practical usage. Travel Sensei fills this gap by combining location marking with a rewarding system, offering credits for visited sights, and celebrating milestones through achievement badges.

- *TripAdvisor* excels in offering user-generated reviews, ratings, and travel planning services. Although it fosters a sense of community through shared insights, it does not encourage consistent interaction on the app or create a competitive environment. Travel Sensei builds on the idea of community by introducing leader-boards, leagues, and a friendship system, motivating users to actively engage and share their achievements with others.

- Gamified platforms like *Strava* focus on specific activities such as cycling and running, but do not provide a wider travel experience. Travel Sensei expands this concept, rewarding users who explore diverse locations and celebrating achievements with badges and leader-boards. Unlike Strava's fitness-centric approach, Travel Sensei emphasizes adventure, social connection, and a sense of community among travellers.

### 1.2.2   Design Constraints

One primary constraint was the need for the application to be functional and easily accessible to users when travelling or moving between locations. This required designing the application to perform reliably in environments with varying network conditions, such as during commutes or in remote areas. To achieve this, the team had to address challenges like maintaining robust security for user data, optimizing data retrieval to work efficiently with potentially limited connectivity, and ensuring scalability for future cross-platform compatibility.

The development timeline was limited, necessitating a focus on creating a Minimum Viable Product (MVP). Some ideas, such as expanding the sight database and advanced gamification features, were postponed for future iterations to ensure that core functionality was delivered on time.

### 1.2.3   Technology Choices

The mobile app was developed using **Android Jetpack Compose** in **Kotlin**, a native solution chosen for its speed of development and platform efficiency compared to cross-platform frameworks such as Flutter. Focusing on a native Android app ensured better performance and a quicker development cycle.

In order to stand out, the application needed a customized map display style. However, we wanted to make sure that whatever mapping provider we choose is not too expensive. We decided to use **MapTiler** as the provider of vector map tiles for our application because it offered greater customization of tiles and a free tier suitable for development. MapTiler also uses open data collected by **OpenMapTiles**, meaning we could decide to host our own tile server later on to save on operational costs.

The backend uses **Spring Boot**, a reliable and efficient framework to create scalable APIs. Spring Boot was selected for its simplicity in development with built-in features and robust performance. Another reason was its seamless compatibility with external services, including AWS, which was critical to achieve an MVP within the constraints. For storing and retrieving relational and geospatial data, **PostgreSQL** with **PostGIS** extension was chosen. Full-text search is enabled by using **Elasticsearch** as our search engine. **PGSync**, an open-source tool, allowed us to have a logical and asynchronous replication from Postgres to Elasticsearch.

For deployment, we have selected **Amazon Web Services** for their popularity, reliability, and accessibility through university-provided resources. AWS also allowed easy integration with Terraform, which was used to manage infrastructure efficiently and ensure scalability.

For admin panel **Python Django** was chosen to develop the admin panel due to its rapid development capabilities and built-in support for administrative tools. Its simplicity allowed the team to quickly implement the core administrative functionality while maintaining flexibility for future enhancements.

### 1.2.4 Data preparation

As our application aims to provide value by providing available points of interest around the world, retrieving, filtering, and processing necessary data is a crucial part for the application to be useful. Due to constrained time, we focused on fetching data for a limited area of Wrocław. We used the **OpenStreetMap** project database, which provides a comprehensive API to retrieve data of interest within specified boundaries and type.

## 1.3 Results

To achieve the desired results, we have created a comprehensive user-interface design mock-up in Figma that followed the Material 3 design language [4]. This mock-up allowed us to better iterate on ideas related to presentation layer and UX before implementing them in code, communicate and discuss changes to what data will need to be available from the back-end, and to follow a consistent look throughout the application.

The UI was then implemented in Jetpack Compose, using a number of custom components, some of which required the use of the lower level Graphics API. The implementation follows Jetpack Compose guidelines on how to manage state and rendering in an optimal way, particularly when it comes to not repeating calculations unless necessary, caching of data and lazily rendering items that may be hidden. To make the UI more fluid, we have created a number of custom animations and transitions. We utilized the MapLibre library to display points of interest on a world map, which is central to our application.

### 1.3.1 Authentication System

An authentication and security system was successfully implemented to provide a seamless and secure user experience. By integrating AWS Cognito, we introduced a robust role-based access control system that supports critical functionalities such as user registration, secure login, email verification, password recovery, and password updates.

The integration of JWT token-based authentication ensured that user sessions were secure and scalable. The system also incorporated the ability to use refresh tokens, minimizing interruptions and enhancing the overall user experience. The benefits for users include faster account setup, enhanced security, and smoother session management. Thanks to that, the user does not need to log in each time they access the application.

The mobile device authentication system also ensures that the user is able to access the application securely and effortlessly, fostering trust and convenience. By using advanced token management, users benefit from uninterrupted sessions and quick re-authentication. This reliability encourages users to spend more time on the app, enhancing engagement and overall satisfaction.

### 1.3.2 Friendship system

The friendship system was designed and implemented to improve user interactions and streamline relationship management within the application. Key functionalities include the ability for users to send, accept, reject, and cancel friendship requests, providing a comprehensive and user-friendly experience. The friendship system is even more attractive to the user due to the many personalisation options available. This personal expression is connected to the user in multiple places in the design, namely the list of friends that contains previews of profiles and the user profile that contains the most personalisation.

To optimize performance, the system incorporates a unique design constraint in which the ID of one user is always smaller than the ID of the other. This ensures that all queries related to friendship status are highly efficient, even as the user base grows. By minimizing the complexity of the checks between two users, the system delivers faster response times to retrieve or update friendship statuses.

### 1.3.3 Milestone system

The milestone system plays a key role in enhancing user engagement by providing a dynamic and rewarding experience. To ensure milestones are processed efficiently without delays, the system uses RabbitMQ streams for seamless asynchronous event handling.

The milestones are another way that a user may use to express themselves, as they can choose milestones to be pinned in the user profile and presented for everyone else to see.

Events related to user achievements are categorized and sent to dedicated streams according to the type of milestone. Each stream processes events through specific subscribers, ensuring accurate and timely updates to users' progress. This approach guarantees smooth performance, even during periods of high activity, without causing long wait times.

By efficiently managing milestones, the system keeps users motivated and engaged, recognizes their achievements promptly, and fosters a sense of accomplishment. The design ensures that the app can handle a growing number of users and milestones without compromising speed or reliability. This foundation not only enriches the user experience, but also supports future expansion of achievement features.

### 1.3.4 Notification system

A notification system leveraging Firebase Cloud Messaging (FCM) was successfully implemented to enhance user engagement and streamline communication. This system ensures that users are promptly informed about important events such as friend requests, rewards (e.g., receiving items), and achievement milestones.

To maintain reliability and security, the notification system integrates dynamic FCM token management. Device tokens are securely associated with users upon sign-in and are automatically removed during sign-out. Invalid tokens, caused by events such as app uninstallation or device changes, are proactively detected and removed from the database. This approach minimizes redundant processing and ensures that notifications reach only valid recipients. Key functionalities include:

- **Real-time event notifications**: activated alerts keep users updated about their achievements and interactions, fostering a sense of engagement and community.

- **Error handling and system efficiency**: invalid tokens are gracefully managed, reducing overhead and maintaining high system performance.

This implementation improves the platform's ability to deliver critical updates reliably, ensuring users remain connected and engaged. By proactively addressing invalid tokens and prioritizing secure delivery, the notification system provides a strong foundation for scalable communication while improving the overall user experience.

### 1.3.5 Map and Sights search

We believe that to create a truly user-friendly application for travelling, the system should be as responsive as possible and provide a smooth experience on the map. When users explore points of interest and search the map, geospatial queries to the database should be efficient and built for scale.

Our aim was to allow users to efficiently explore the desired region, search using text, and retrieve the destination sights. In addition, the system should notify the user if the user is within the boundaries of certain sightseeing to eventually be able to see the history of visits.

From the above-mentioned user needs, we identified two main query patterns:

- **Range search**. Range queries answer the questions, such as e.g. *Find all points within 5 km of the current position*. This query addresses the problem of finding nearby points or points in certain boundaries that are interesting to users.

- **Geofencing**. Geofencing queries answer the question e.g. *Find all sightseeing within the boundaries of which user is currently in*. This query addresses the problem of checking and notifying the users about visiting the place. Geofencing queries are popular in many applications, when there is a need for detection when certain objects enter the desired or unwanted zone (e.g., fleet and freight management) [8].

Storing points of interest in traditional relational database systems (RDBMS) is difficult because the nature of the data is multidimensional. Spatial Database Management Systems (SDBMS) and spatial indexes are commonly used to solve geospatial search and storage problems.

Two geospatial search solutions, PostGIS and its R-Tree [6] index structure, and more recently developed hexagonal grid H3 [9], were investigated against our query patterns. It should be noted that H3 is not a database index, but a tool that can be used for a performant search.

We wanted to ensure that users receive minimum false positive visit notifications; therefore, polygons of points of interest were used for the PostGIS solution (Figure 1), and high resolution hexagon tiles were used for the H3 solution (Figure 2).

Although the geofencing problem can be solved in different ways, such as using the Android API [3] in the context of a mobile application, it has certain limitations and tends to consume significant CPU and

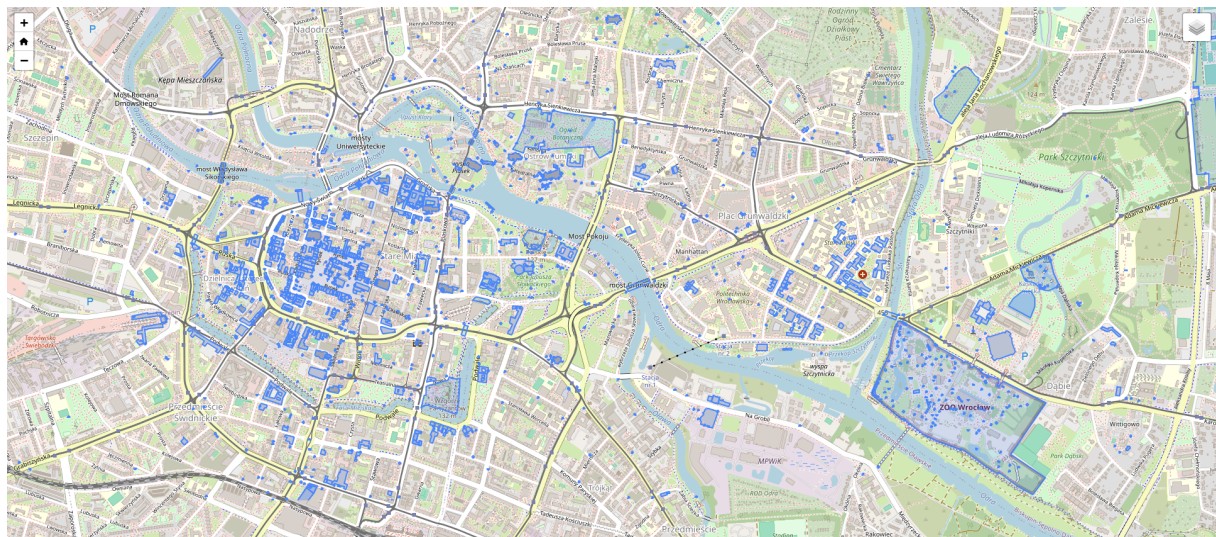

Figure 1: Polygon representation of sights in our PostGIS database in Wrocław

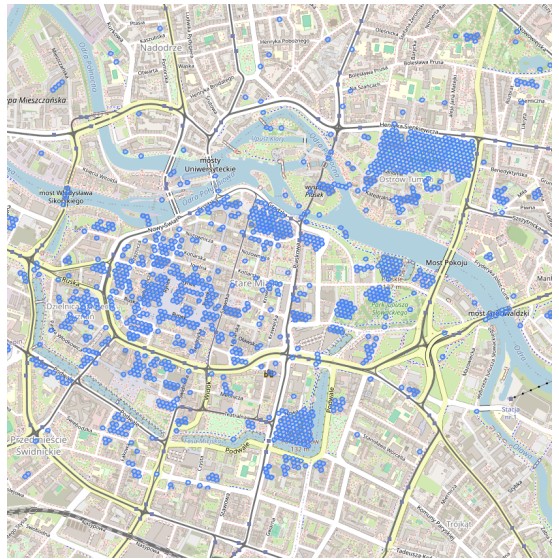
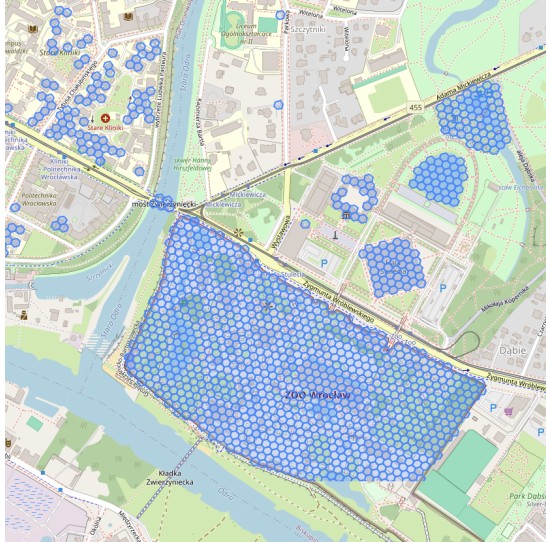

Figure 2: H3 representation of sights in our PostGIS database in Rynek and Zoo, Wrocław

battery resources. Therefore, we concentrated on solving this problem at the back-end and database level, using spatial data structures and algorithms to offload the computational burden of the client device.

Since both R-Tree and H3 emerged as strong candidates for most geospatial use cases, we needed to determine which option would perform better for our specific requirements. In order to compare performance, points-of-interest data was loaded from Wrocaw and other cities, equalizing approximately 17,500 points, which would be enough for the initial phase of the application and appeared to be sufficient for query planner to use index scans over sequential scans. Among the numerous database benchmarking tools, pgbench [5] was chosen as the primary candidate. The database was compared for geofencing query patterns using pgbench, which provides output metrics such as latency and transaction throughput, and different configuration options.

As shown in Figure 3, the benchmarking revealed a significant performance difference between two approaches for geofencing when tested with a varying number of concurrent clients (10, 30, and 60). The H3 method consistently demonstrated superior transaction throughput across all concurrency levels, resulting in 20- 25% better performance, achieving 42,000 TPS versus 34,000 TPS in 10 clients, with the advantage maintained at higher concurrency levels. Similarly, in terms of latency, H3 outperformed R-Tree when the number of clients was increasing.

It is crucial to note that these results specifically pertain to geofencing query patterns, which are particularly suited to H3's hierarchical hexagonal grid structure. Range query performance analysis, while crucial for other aspects of our applications, such as nearby point discovery, was not evaluated

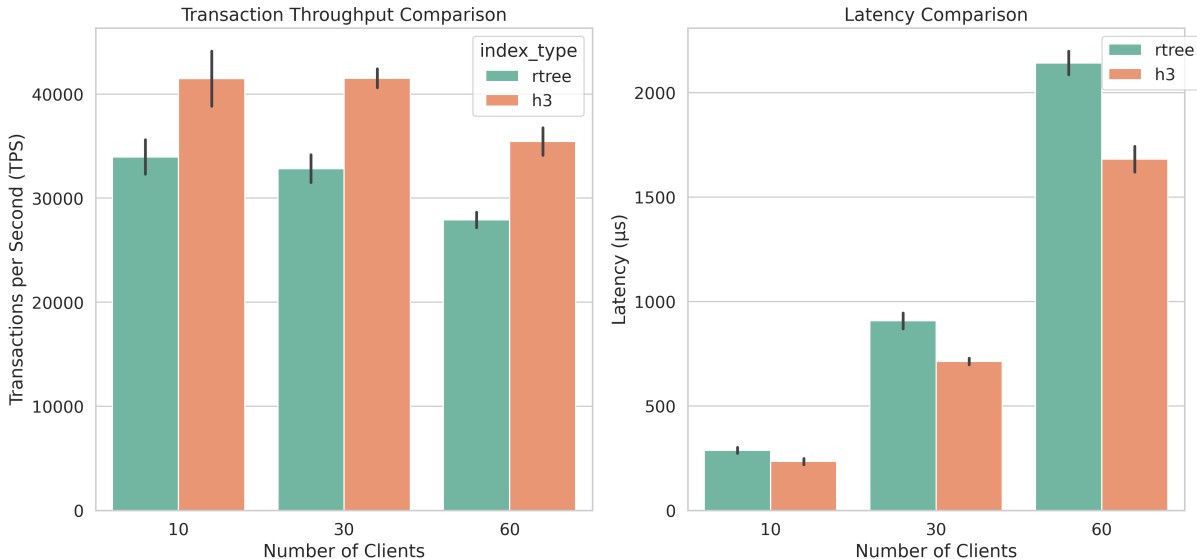

Figure 3: Comparison of R-Tree index vs H3 index for geofencing

due to time constraints and remains an area for future investigation.

This performance analysis guided our technical decisions to implement the geofencing part of the application, when the choice of H3 will significantly improve the user experience and responsiveness of the application.

**On the mobile part**, the implementation serves as a robust complement to the back-end geospatial architecture. A dynamic data loading strategy has been adopted, using map boundaries and pagination to selectively retrieve points of interest as required. The following notable features characterize the mobile solution:

- **Dynamic bounds-based fetching**: this method involves retrieving data specifically for the area currently displayed on the map. By limiting data downloads to what is relevant, the system works in a "on-demand" way, thereby facilitating a fluid user experience as users engage exclusively with visible sights.

- **Pagination for scalability**: the architectural design incorporates pagination to load data in manageable increments. This technique allows the system to effectively accommodate substantial datasets while preventing potential overload on both back-end systems and mobile devices.

- **Local caching with Room**: the implementation includes a local caching mechanism that stores previously accessed data. This feature significantly diminishes the need for repeated API calls and supports offline access to areas that users have recently explored

- **Seamless synchronization**: remote mediator oversees the synchronization of the local database with back-end data. This ensures that users can access the most current information without any degradation in performance.

This comprehensive approach not only enhances user engagement, but also aligns with the overarching objectives of the system by ensuring responsiveness, scalability, and efficient use of resources.

### 1.3.6   Custom designed elements

The uniqueness of our application was enhanced by creating originally designed assets for profile customisation elements and milestone badges. The following elements for custom profile were created: user avatars, suitable frames designed specifically to match each avatar type, as well as more generic frames, and backgrounds for profile header. The milestone badges were designed to precisely illustrate the nature of each concrete achievement to be easily recognized by the users. The Figma tool was used for their creation. These designs were made to look attractive and pleasant to catch the user's attention. When designing these elements, it was important to keep them consistent. In the end, no matter which

set of profile customisations is chosen, it will always match any milestones present on that specific profile.

From a technical point of view, all assets were manually designed in the SVG (Scalable Vector Graphics) format, chosen specifically for its versatility and technical advantages. Such a format choice allows for flawless scaling across devices with varying screen sizes and resolutions, from standard displays to high-DPI screens. Because they have a vector-based nature, SVG assets are very lightweight and do not occupy a lot of device's memory, which allows optimizing the application's performance.

## 2 CONCLUSIONS

The Travel Sensei project successfully achieved its goal of transforming travel into an engaging, social, and gamified experience. By incorporating features such as personalised achievements, a friendship system, and competitive leader-boards, the application fosters exploration, builds community connections, and enhances user engagement. Key technical accomplishments include a robust back-end powered by Spring Boot, PostgreSQL with PostGIS, and AWS for scalability, as well as an intuitive UI implemented with Jetpack Compose.

The most significant success of the project lies in its innovative use of gamification to promote mental well-being and foster a sense of adventure, making Travel Sensei a standout platform in the travel app market. Its scalable design and expansion plans ensure long-term growth and monetization potential, promising users a rewarding travel experience throughout the world.

### 2.1 Future directions

Looking ahead, Travel Sensei has a strong potential for growth and monetization. Future enhancements include expanding the gamification elements, introducing new features, and partnering with local businesses to provide exclusive integrations and discounts, further enriching the user experience:

- **Expansion of geographical coverage**: currently focused on Wrocław, the platform will be expanded to cover additional regions, offering a global network of destinations. This will enable users to use the app during travels anywhere in the world, significantly increasing its utility and reach.

- **User-driven content integration**: a new feature will be introduced that will allow users to discover and suggest new locations. These user-submitted places, once verified, will be added to the app, ensuring an ever-growing and dynamic database that reflects real-time travel trends and hidden gems.

- **Tangible rewards for credits**: the current virtual rewards system is to be expanded to include tangible rewards that users can redeem using their earned credits. Examples include travel-related items, gift cards, or vouchers, creating additional incentives for active engagement.

- **Advanced recommendation system**: a recommendation engine powered by AI and user data analytics is also in the future plans. This system will provide personalised suggestions for nearby attractions, dining options, and events based on individual preferences and travel history, enhancing the user experience.

### 2.2 Acknowledgments

We would like to thank the teams behind the MapLibre Native library [1] and the related Compose wrapper library [7] for making a high-quality open-source vector map display library, authors of the open-source tool Pgsync [10] for creating a comprehensive tool for synchronization, and OpenStreetMap [2] for the project that creates and distributes free geographic data for the world.

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
