# OpenReview forum: "Travel Sensei - Turning Travel into Adventure"
_pwr.edu.pl/Wrocław_University_of_Science_and_Technology/2024/ZPI_Day — Wrocław University of Science and Technology 2024 ZPI Day Submission_

### Official Review · Reviewer_6eNr · 2024-12-06
**Recenzja projektu Travel Sensei**

**Confidence:** 5
**Significance Of Results:** 5
**Overall Quality:** 5

**Compliance With Template:**

5: Very High Quality – The article contains all the required sections, which are written in a very detailed, clear, and error-free manner. The structure is professional and meets expectations, and the content adheres to the highest substantive and formal standards.

**Description Of Results:**

5: Very High Quality – The results are described in detail, clearly and comprehensively, supported by thorough evaluation, analysis, and convincing usage examples. The description meets the highest substantive standards.

**Feedback On Consistency:**

Abstrakt napisany jest w sposób spójny i logiczny. Problem został jasno sprecyzowany. Wyniki przedstawiono w sposób bardzo szczegółowy.

**Potential For Development:**

Autorzy zidentyfikowali cztery potencjalne kierunki rozwoju, które zdecydowanie mogą uatrakcyjnić projektowane rozwiązanie.

**Project Nature Evaluation:**

Układ abstraktu jest właściwy, a cel jasno sprecyzowany. Cel został osiągnięty. Narzędzia i technologie są adekwatne. Uważam, że projekt spełnia wymogi stawiane projektom inżynierskim.

**Technical Language Precision:**

5: Very High Quality – The language is entirely appropriate for a technical report. All terms are used correctly and precisely, and the style is professional, clear, and coherent, without any errors or ambiguities.

---

### Official Review · Reviewer_aRnD · 2024-12-06
**An innovative app with significant potential to enhance societal well-being by addressing the unwillingness to travel.**

**Confidence:** 4
**Significance Of Results:** 5
**Overall Quality:** 5

**Compliance With Template:**

5: Very High Quality – The article contains all the required sections, which are written in a very detailed, clear, and error-free manner. The structure is professional and meets expectations, and the content adheres to the highest substantive and formal standards.

**Description Of Results:**

5: Very High Quality – The results are described in detail, clearly and comprehensively, supported by thorough evaluation, analysis, and convincing usage examples. The description meets the highest substantive standards.

**Feedback On Consistency:**

The article is generally consistent in presenting the problem analysis, results, and conclusions. The flow between these sections is logical, and the conclusions align with the results presented. Considering several different solutions chosen to achieve the application's goals, the division into chapters is logical and coherent.

**Potential For Development:**

The article does suggest possibilities for further work or practical applications of the results. It provides a brief outline of potential future developments, although additional details on how the project could be monetized and the potential business model(s) that could support its growth, would provide a more thorough outlook on future potential.

**Project Nature Evaluation:**

The project exhibits strong characteristics of an engineering work. It involves technical methods, addresses a clear problem, and provides practical technological solutions. The results demonstrate a clear application in a real-world context, which aligns with the expectations for an engineering project.

**Technical Language Precision:**

5: Very High Quality – The language is entirely appropriate for a technical report. All terms are used correctly and precisely, and the style is professional, clear, and coherent, without any errors or ambiguities.

---

### Official Review · Reviewer_uSJp · 2024-12-08
**Great solution, focused on the problem and not buzzwords**

**Confidence:** 5
**Significance Of Results:** 5
**Overall Quality:** 5

**Compliance With Template:**

5: Very High Quality – The article contains all the required sections, which are written in a very detailed, clear, and error-free manner. The structure is professional and meets expectations, and the content adheres to the highest substantive and formal standards.

**Description Of Results:**

5: Very High Quality – The results are described in detail, clearly and comprehensively, supported by thorough evaluation, analysis, and convincing usage examples. The description meets the highest substantive standards.

**Feedback On Consistency:**

The content of the submission is consistent and logical. All requested parts of the document are present and are connected in a logical manner. The authors present quite a detailed introduction to the subject, then build on it for the rest of the paper. The paper is focused on a single topic:
providing valuable service to the customer, and not on buzzwords.

**Potential For Development:**

Project has a huge potential for development. The authors discussed some of them, again, as was in the previous sections on problems. Also, the discussion of comparison results, between original and developed way of storing information, show other directions to improve and extend the project.

**Project Nature Evaluation:**

The paper presents a very interesting technical solution. The authors focused not only on putting existing services together with some glue code. Instead, they solved the problem of storing geospatial data because existing services did not offer the approach suitable for their application. The paper contains also a comparison of the achieved results to the base values. To summarize, the authors discussed limitations and possible extensions of achieved results.
The team focused on solving the problem that is the key to the right software architecture.
The final result is a mature project showing the most important elements of engineering work.

**Technical Language Precision:**

5: Very High Quality – The language is entirely appropriate for a technical report. All terms are used correctly and precisely, and the style is professional, clear, and coherent, without any errors or ambiguities.

---

### Decision · Program_Chairs · 2024-12-10

Accept (Poster)